# TCP Transcription Factors in Pineapple: Genome-Wide Characterization and Expression Profile Analysis during Flower and Fruit Development

Ziqiong Li [1], Yanwei Ouyang [1], Xiaolu Pan [1], Xiaohan Zhang [1], Lei Zhao [1], Can Wang [1], Rui Xu [1], Hongna Zhang [1,*] and Yongzan Wei [2,*]

1 Sanya Nanfan Research Institute of Hainan University, Hainan University, Sanya 572025, China; liziqiong2021@163.com (Z.L.); oyyw124@163.com (Y.O.); pxlat4820@163.com (X.P.); zhangxiaohan1205@163.com (X.Z.); 21220951310191@hainanu.edu.cn (L.Z.); canw0131@163.com (C.W.); xr123000z@163.com (R.X.)

2 Key Laboratory of Biology and Genetic Resources of Tropical Crops, Ministry of Agriculture, Institute of Tropical Bioscience and Biotechnology, Chinese Academy of Tropical Agricultural Sciences, Haikou 571101, China

* Correspondence: 994357@hainanu.edu.cn (H.Z.); wyz4626@163.com (Y.W.)

**Abstract:** TEOSINTE-BRANCHED1/CYCLOIDEA/PCF (TCP) transcription factors contain specific a basic helix–loop–helix structure, which is a significant factor in the regulation of plant growth and development. TCP has been studied in several species, but no pineapple TCP has been reported to date. Whether they are involved in the development of the flower and fruit in the pineapple remains unclear. In this study, nine non-redundant pineapple TCPs (AcTCPs) were identified. Chromosomal localization, phylogenetics, gene structure, motifs, multiple-sequence alignment, and covariance on AcTCP family members were analyzed. Analysis of promoter *cis*-acting elements illustrated that the *AcTCP* gene may be mainly co-regulated by light signal and multiple hormone signals. Analysis of expression characteristics showed a significant increase in *AcTCP5* expression at 12 h after ethylene treatment, and significantly higher levels of *AcTCP8* and *AcTCP9* expression in the pistil than in other floral organs. Meanwhile, the *AcTCP4, AcTCP5, AcTCP6, AcTCP7*, and *AcTCP9* expression levels were downregulated at later stages of fruit development. Transcription factors that may interact with TCP protein in the regulation of flower and fruit development are screened by the protein interaction prediction network, AcTCP5 interacts with AcSPL16, and AcTCP8 interacts with AcFT5 and AcFT6 proteins, verified by Y2H experiments. These findings provide a basis for further exploration of the molecular mechanisms and function of the *AcTCP* gene in flower and fruit development.

**Keywords:** TCP transcription factors; pineapple (*Ananas comosus* (L.) Merr.); expression profiling; flower development; fruit development; protein interaction

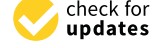



## 1. Introduction

The TEOSINTE-BRANCHED1/CYCLOIDEA/PCF (TCP) gene family encodes proteins that are plant-specific and play an essential role in the modulation of plant growth. The TCP transcription factor was first described to have originated from three special gene family members, TB1 (TEOSINTE BRANCHED1) in maize, which can control apical dominance in maize [1]; CYCLOIDEA (CYC), which can regulate the bilateral symmetry of Antirrhinum flowers [2]; and proliferating cell factor 1 and 2 (PCF1 and PCF2) in rice, which are involved in the meristem-specific expression of the rice PCNA (proliferating cell nuclear antigen) gene [3]. TCP proteins have a highly conserved basic helix–loop–helix (bHLH) structure consisting of 59 residues, called the TCP structural domain [4]. According to the phylogenetic analysis and structural characteristics of the TCP gene family, they could be divided into two classes, class I (TCP-P) and class II (TCP-C), with class I containing

the PCF subfamily and class II containing the CIN and CYC/TB1 subfamilies. Obvious differences exist between these two classes of proteins, with the basic structural domain of class I lacking four amino acid residues compared with that of class II [5]. In addition, the PCF subfamily encodes proteins that bind to the GGNCCCAC element, and the class II TCP family encodes proteins that bind to G(T/C) GGNCCC element, thereby regulating downstream gene expression [6].

The TCP family is ubiquitous in higher plants. They have 23 members in *Arabidopsis* [7], 21 members in *Ziziphus jujuba* [8], 18 members in grape [9], 18 members in strawberry [10], and 29 members in maize [11]. As a conserved and widespread transcription factor family in plants, TCP has evolved various methods to accurately regulate its downstream target genes in different plants, and its gene expression has been strictly regulated [12]. TCP transcription factors could regulate many biological processes, such as flower development, fruit development, leaf development, and plant hormone signal transduction [13–24]. Some *TCP* members are regulated by microRNA319 [25–28]. TCP proteins regulate flower development. *Arabidopsis* TCP5/13/17 (class II TCP protein) interacts with FD proteins, integrates into the FT-FD module, promotes *AP1* expression, and positively regulates flowering in an AP1-dependent manner [13]. In *Arabidopsis*, AtTCP4 has the function of regulating flowering time, *TCP4* deletion mutants flower later than the wild type, and *TCP4* overexpression mutants (*TCP4: VP16-C*) flower earlier than the wild type [14]. AtTCP7 interacts with NF-Y to activate the transcriptional expression of the flowering integration gene *SOC1*, which makes *Arabidopsis* flower earlier [15]. TCP5 restricted petal growth in a dose-dependent manner. The *TCP5* deletion mutants in *Arabidopsis* had wider petals than those in the wild type, and the petal width of *TCP5* overexpression plants was significantly reduced [16]. In *Chrysanthemum morifolium*, CmCYC2b-CmCYC2d, CmCYC2b-CmCYC2e, and CmCYC2c-CmCYC2d interact to form heterodimeric complexes, and CmCYC2c has the ability to interact with the promoter region of *ClCYC2f*, thereby controlling the development of flower symmetry [17]. TCP protein can promote or inhibit fruit ripening during fruit development. In woodland strawberry, FvTCP9 modulates the expression of genes associated with ABA signaling (*FvNCED1*, *FvPYR1*, *FvSnRK2*, and *FvABI5*) and interacts with FaMYC1 to regulate anthocyanin biosynthesis, thereby promoting fruit ripening [18]. In banana, MaTCP20 and MaTCP5 promote the expression of *MaXTH10/11* to soften the fruit, and MaTCP19 inhibits the expression of *XTH10/11*. In addition, MaTCP20 interacts with MaTCP5 to form a dimer, which could promote the transcription of *XTH10/11*, and the combination of MaTCP20 and MaTCP19 offsets its promotion of *XTH10/11* transcription [19]. Tomato TCP proteins can interact with other proteins of this family to form dimers. In addition, the expression of *TCP12*, *TCP15*, and *TCP18* genes is regulated by RIN (ripening inhibitor), CNR (colorless non-ripening), and SlAP2a (APETALA2a) proteins, which play an important role in fruit ripening, and TCP proteins can also bind to promoters of members of this family to co-regulate tomato fruit formation [20].

Pineapples are perennial herbaceous fruit trees under the family Bromeliaceae. It has important processing and fresh food value [29]. Considering the significant contribution of the TCP family to flower and fruit development, and the lack of previous reports on the TCP family in pineapples, a comprehensive survey of the TCP family in pineapples was conducted in this study. Nine non-redundant *TCP* genes were identified and systematically analyzed for chromosome mapping, phylogenetics, gene structure and motif, multiple-sequence alignment, covariance, and promoter *cis*-acting elements. The expression profiles of the *AcTCP* gene at various stages of flower and fruit development were also analyzed further, and proteins that interact with TCP were predicted and validated, laying the foundation for functional validation of the pineapple gene.

## 2. Materials and Methods

### 2.1. Plant Growth and Flower Induction

The pineapple plants (*Ananas comosus* L. cv. Comte de Paris) were obtained from the pineapple planter base of Zhanjiang, China (21°10′2″ N, 110°16′34″ E). Homogenous plants with 25 leaves 30 cm long were selected for perfusion treatment to induce flowering. The treatment group was perfused with 30 mL of 400 mg/L ethephon and the plants were treated with an equal volume of water as the control group. The shoot apices of pineapple were also collected at six time points: 0 h, 12 h, 1 d, 7 d, 14 d, and 21 d after treatment. The petals, ovary, stamens, sepals, and stylet were simultaneously collected in full bloom. Fruits were also collected at 6 time points, 2, 4, 6, 8, 10, and 11 weeks after treatment. Three independent biological replications should be inclusive of every sample.

### 2.2. Identification of TCP Gene Family in Pineapple

Genome and proteome sequences were downloaded from the pineapple genome database (http://pineapple.zhangjisenlab.cn/pineapple/html/index.html, accessed on 2 June 2022) [30]. Twenty-four AtTCP transcription factor protein sequences were downloaded from the Arabidopsis database (https://www.arabidopsis.org/, accessed on 2 June 2022) [31]. The *AcTCP* candidate gene was obtained by using AtTCP protein sequences and BLAST to find the most similar sequences from the pineapple genome database. All of the candidate genes were further examined by the NCBI Conserved Domain Database (https://www.ncbi.nlm.nih.gov/Structure/cdd/cdd.shtml, accessed on 1 August 2022) to confirm that they contain the TCP domain structure.

Prediction of the molecular weight (MW), isoelectric point (PI), Aliphatic index, and grand average of hydropathicity (GRAVY) of AcTCP transcription factors using the online site (https://www.expasy.org/, accessed on 8 August 2022) [32]. The CELLO v.2.5 website (http://cello.life.nctu.edu.tw/, accessed on 9 August 2022) was used to predict the subcellular location of AcTCP proteins [33].

### 2.3. Chromosomal Distribution and Evolutionary Analysis of the AcTCP Gene Family

The localization information of *AcTCP* gene chromosomes was obtained from the pineapple genome database, and the chromosome localization information map was completed using MG2C (http://mg2c.iask.in/mg2c_v2.1/, accessed on 10 August 2022) [34]. All TCP protein sequences of maize, grape, and *Arabidopsis* were downloaded from the UniPort database (https://www.uniprot.org/, accessed on 12 August 2022), and phylogenetic trees were constructed by the maximum likelihood (ML) method using MEGA 7.0 software with 1000 bootstrap replicates [35].

### 2.4. Conserved Motifs and Gene Structure Analysis of AcTCP Gene Family

The conserved motifs of AcTCP were analyzed through the MEME website [36] (https://meme-suite.org/meme/tools/meme, accessed on 15 August 2022), with parameters set to classic mode, an unlimited number of repetitions, and a maximum number of motifs of eight. Intron structure analysis of AcTCP was performed based on genome-wide GFF annotation files. Visual analysis was performed using TBtools software [37].

### 2.5. Gene Duplication and Syntenic Analysis of the Pineapple AcTCP Gene Family

The genome sequences of maize, rice, grape, and Arabidopsis were downloaded from the Ensembl Plants database (https://plants.ensembl.org/index.html, accessed on 20 August 2022), the gene duplication events of pineapple with each of the four species were plotted using TBtools software, and the TBtools' dual synteny plot tool was used to complete the covariance analysis plots.

### 2.6. Cis-Acting Regulatory Elements in the Promoter and Protein Interaction Prediction Analysis

The first 2000 bp sequences of the transcriptional start site of the nine AcTCP genes were submitted to PlantCARE (https://bioinformatics.psb.ugent.be/webtools/plantcare/

html/, accessed on 20 September 2022) [38]. Prediction of promoter *cis*-acting elements was performed. Interaction predictions for nine AcTCP genes were performed using the protein interaction prediction online website (STRING https://cn.string-db.org/, accessed on 25 September 2022) [39].

### 2.7. RNA Isolation, qRT-PCR Analysis of AcTCPs in Different Issues, and the Stage of Development

Each sample was weighed with 500 mg of pineapple tissue for total RNA extraction using the RNA extraction kit (Huayueyang, Beijing, China). RNA quality and concentration were detected using NanoDrop™ One/OneC Spectrophotometer (Thermo Fisher Scientific, Waltham, MA, USA) and 1.5% agarose gels. The concentrations of obtained total RNAs were 150–500 ng/μL. Then, the first-strand cDNAs were synthesized from 1 μg of total RNA via the Revert Aid First-Strand cDNA Synthesis Kit (Thermo Fisher Scientific, USA).

Primers for pineapple *AcTCP* genes were designed using BatchPrimer3 online (Table S1). For qRT-PCR analysis, the *AcActin* gene was used as a reference gene. The qRT-PCR was performed using SYBR-green fluorescence with Light Cycler 480 II (Roche, Basel, Switzerland). The reaction mechanism was performed in a 10 μL volume, including 5 μL of 2 × SYBR Green PCR Master Mix (Applied Biosystems, Waltham, MA, USA), 1 μL of cDNA, and 1 μL of primers. The expression level of *AcTCP* genes was calculated by the $2^{-\Delta\Delta Ct}$ [40] method, and three biological replications were used in all the experiments.

### 2.8. Statistical Analysis

Statistically significant differences were determined using the LSD test. The mean ± standard error (SE) of three replicates is presented. Data were analyzed using SPSS Statistics 25.0 software. Lowercase letters indicate significant differences at $p \leq 0.01$.

### 2.9. Yeast Two-Hybrid Analysis

In order to verify the interaction between TCP protein with SPL and FT proteins, the online website (https://blast.ncbi.nlm.nih.gov/Blast.cgi, accessed on 20 January 2023) was used to analyze the protein sequence with the highest homology to SPL9 and FT in the pineapple genome. The full-length cds of AcSPL4, AcSPL16, AcSPL17, FT2, FT5, and FT6 were cloned into PGBKT7 vector, and the full-length AcTCP5 and AcTCP8 were cloned into PGADT7 vector. The BD vector and AD vector were co-transformed into a yeast-competent AH109 strain and coated on SD/Trp/leu. Healthy yeast cells were selected and coated on the selection medium SD/Trp/Leu/His/Ade/3-AT (5/10 mM). After 48–72 h of culture, X-a-Gal was added drop by drop to further verify whether TCP protein interacted with SPL and FT proteins.

## 3. Results

### 3.1. Identification and Classification of TCP Gene Family in Pineapple

*TCP* genes of pineapples were identified by BLAST search. The TCP proteins in *Arabidopsis* were employed as a query object to search for homologous sequences of the pineapple genome by using TBtools, and nine non-redundant *TCP* genes of pineapple were obtained. They were named *AcTCP1-AcTCP9* based on the position of the chromosomal mapping. Details of the physical and chemical properties of *AcTCP* genes are listed in Table 1. The molecular length and weight of the nine AcTCP proteins ranged between 253 (AcTCP9) and 532 (AcTCP8) amino acid residues, with relative molecular masses ranging from 25.54 kDa (AcTCP9) to 57.7 kDa (AcTCP8). Meanwhile, the theoretical isoelectric point (pI) values of AcTCP ranged from 5.35 to 10.35. Six of the members had pI < 7. Thus, presumably, the family members are mostly acidic. The lipid index ranged from 52.44 to 75.43, all of which were greater than 40, so the protein was possibly unstable. The values of GRAVY were all less than 0, so the proteins are presumably hydrophilic [41]. The subcellular localization prediction analysis indicated that all nine *AcTCPs* exhibited nuclear expression.

**Table 1.** TCP gene family identification and protein properties analysis in pineapple.

| Gene Name | Gene ID | Maximum ORF Length (bp) | Amino Acid Length (aa) | MW (kDa) | Theoretical pI | Aliphatic Index | GRAVY | Predicted Subcellular Location |
|---|---|---|---|---|---|---|---|---|
| *AcTCP1* | Aco012417.1 | 807 | 269 | 30.21 | 7.89 | 75.43 | −0.607 | Nuclear |
| *AcTCP2* | Aco024489.1 | 1062 | 354 | 39.43 | 9.13 | 69.77 | −0.503 | Nuclear |
| *AcTCP3* | Aco006659.1 | 1014 | 338 | 34.37 | 6.59 | 62.07 | −0.51 | Nuclear |
| *AcTCP4* | Aco002292.1 | 1020 | 340 | 37.89 | 5.89 | 71.38 | −0.694 | Nuclear |
| *AcTCP5* | Aco003020.1 | 963 | 321 | 35.66 | 6.27 | 60.59 | −0.795 | Nuclear |
| *AcTCP6* | Aco021664.1 | 1065 | 355 | 37.42 | 5.47 | 63.94 | −0.646 | Nuclear |
| *AcTCP7* | Aco015741.1 | 945 | 315 | 32.55 | 6.17 | 65.05 | −0.499 | Nuclear |
| *AcTCP8* | Aco010666.1 | 1596 | 532 | 57.70 | 5.35 | 52.44 | −0.848 | Nuclear |
| *AcTCP9* | Aco010326.1 | 759 | 253 | 25.54 | 10.35 | 67.83 | −0.255 | Nuclear |

### 3.2. Chromosomal Mapping of AcTCP Gene Family

The chromosome location of *AcTCPs* was analyzed in accordance with the genome annotation information of pineapple, and the MG2C website was used to draw the chromosomal map. The findings revealed that the nine *AcTCPs* were located on seven distinct chromosomes. Among them, *AcTCP1*, *AcTCP2*, and *AcTCP3* were located at LG01, *AcTCP4* was at LG04, *AcTCP5* was at LG06, *AcTCP6* was at LG07, *AcTCP7* was at LG09, *AcTCP8* was at LG10, and *AcTCP9* was at LG25 (Figure 1). The results indicated that *AcTCPs* were unevenly distributed on seven chromosomes of pineapple.

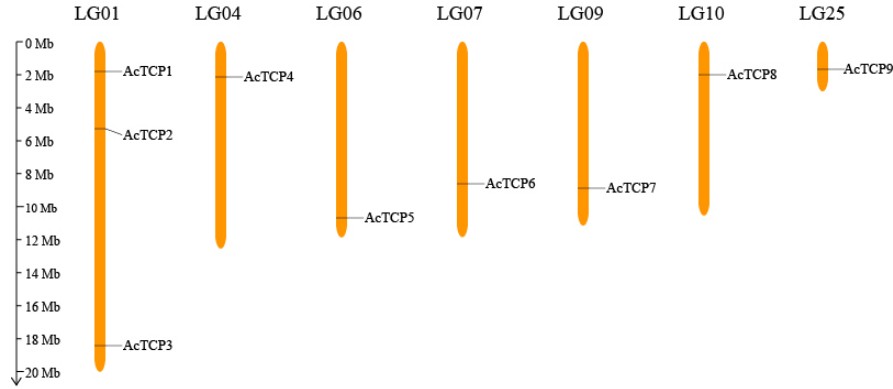

**Figure 1.** Locations of *AcTCP* genes in pineapple chromosomes. Nine *AcTCP* genes were distributed on seven chromosomes. The vertical bar represents the chromosomes in the pineapple genome. The scale on the left indicates the length of each chromosome.

### 3.3. Phylogenetic Analysis

To explore the evolutionary relationship between *TCP* genes in pineapple and other plants such as *Arabidopsis*, grape, and maize, an evolutionary tree was constructed (Figure 2). The results revealed that these genes could be categorized into two major classes, class I and class II, which were further divided into subfamilies, including PCF, CIN, and CYC/TB1. *AcTCP3*, *AcTCP6*, *AcTCP7*, and *AcTCP9* belonged to the PCF subfamily; *AcTCP1*, *AcTCP4*, and *AcTCP8* belonged to the CIN subfamily; and *AcTCP2 and AcTCP5* belonged to the CYC/TB1 subfamily. *TCP* genes belonging to the same subfamily may have higher sequence similarity and thus have similar functions.

### 3.4. AcTCP Gene Structure, Motif Analysis, and Multiple-Sequence Alignment of AcTCP Protein Sequences

Gene structure and motif analyses were conducted to gain deeper insights into the evolutionary relationships and structural characteristics of TCP proteins in pineapple. The protein sequences of nine AcTCPs were utilized to construct an evolutionary tree,

which resulted in their classification into three distinct groups based on their evolutionary relationships. The eight conserved motifs with the highest frequencies among the nine TCP proteins of pineapple were analyzed using the MEME website (Table S2). All nine AcTCPs were found to contain motif 1, which is an arginine-rich R domain. In addition, all class 1 subfamilies contain motif 3, all class 2 subfamilies contain motif 6, and the CYC/TB1 subfamily contains motif 7 (Figure 3A).

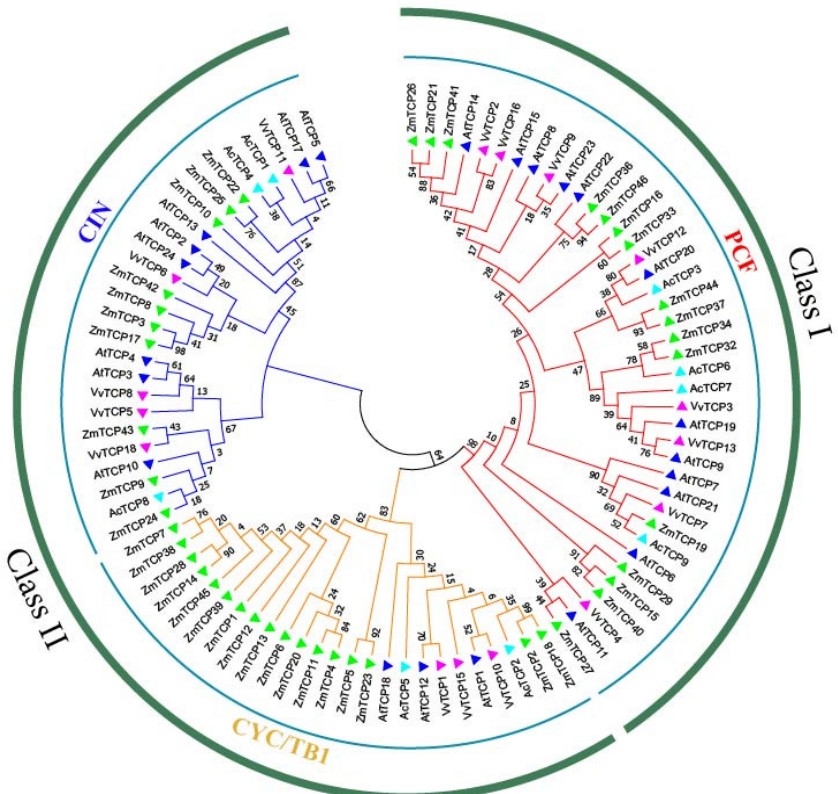

**Figure 2.** Phylogenetic analysis of *TCP* family among pineapple, *Arabidopsis*, grape, and maize. The subtrees are depicted with colored branches to represent distinct TCP subgroups.

The analysis of gene structure revealed that *AcTCP4* and *AcTCP8* contain three exons, *AcTCP9* contains two exons, and the other members contain only one exon; *AcTCP9* contains one intron, and *AcTCP4* and *AcTCP8* contain two introns (Figure 3B). The *AcTCP* genes in the same subfamily showed similar exon patterns. The lack of introns in the members of the CYC/TB1 subfamily and the smallest number of members of this subfamily are inconsistent with the pattern found in previous studies; that is, genes lacking introns evolve rapidly through gene duplication events [42], may be due to the small number of genes that are not regular.

The multiple-sequence alignment analysis of AcTCPs revealed that all nine TCP family members contain intact bHLH structural domains (Figure 3C), with 21 amino acid residues on the basic conserved structural domain and a putative nuclear localization signal. The helix region has alternating conserved hydrophobic residues, partially conserved hydrophilic residues, and an LXLL motif. The loop region connecting the two helices conserves glycine, aspartate, and serine residues. Moreover, proline is present in CYC/TB1 and PCF. A total of 13 highly conserved amino acids were found in the TCP structural domain, including 7 in the basic region and 6 in the HLH region.

*3.5. AcTCP Gene Covariance Analysis*

Intra-species covariance analysis was conducted to investigate the occurrence of gene duplication events in the *AcTCP* gene family, and the results demonstrated that no

fragmental or tandem duplication events occurred. Between-species covariance analysis of pineapple with maize, rice, *Arabidopsis*, and grape revealed that five *AcTCPs* generated eight gene pairs between pineapple and maize; five *AcTCPs* generated nine gene pairs between pineapple and rice; five *AcTCPs* generated six gene pairs between pineapple and *Arabidopsis*; and three *AcTCPs* generated five gene pairs between pineapple and grape (Figure 4). The results indicated that pineapple was more closely related to monocotyledonous plants than to dicotyledonous plants.

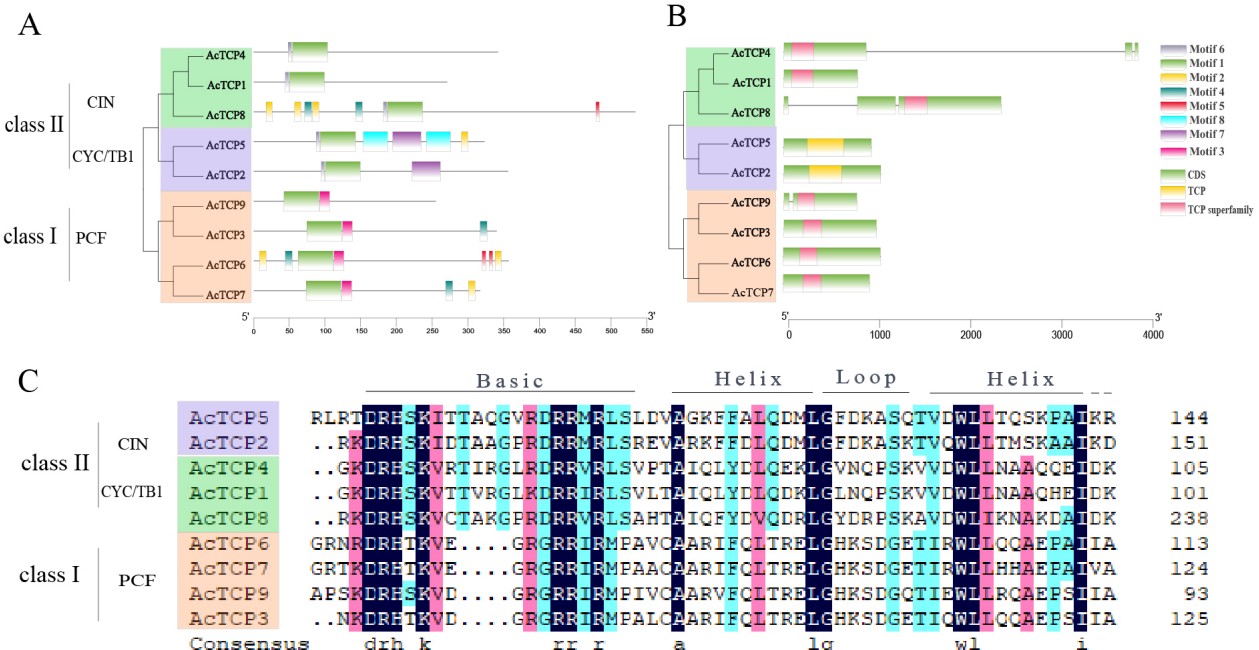

**Figure 3.** Gene structure, conserved motifs, and multiple-sequence alignment of TCP family in pineapple; (**A**) Distribution of predicted motifs in AcTCP proteins; (**B**) Exon–intron gene structure of *AcTCP* family; (**C**) Multiple-sequence alignment of TCP domain in pineapple.

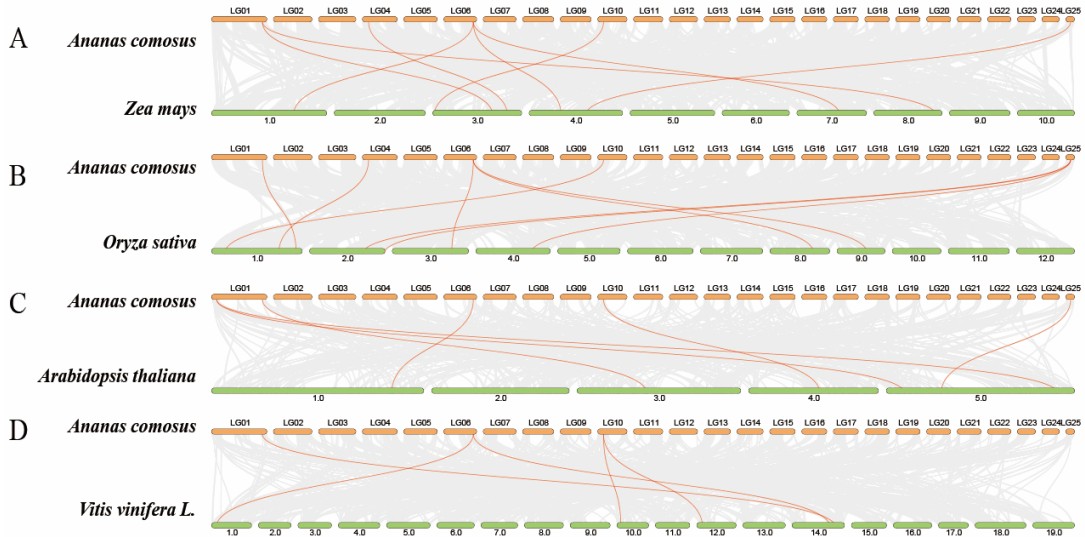

**Figure 4.** *AcTCP* gene covariance analysis. Covariance analysis of *TCP* genes between pineapple and other plants (maize, rice, *Arabidopsis*, and grape). The red line represents the homologous gene pair: (**A**) Covariance analysis of TCP genes between pineapple and maize; (**B**) Covariance analysis of TCP genes between pineapple and rice; (**C**) Covariance analysis of TCP genes between pineapple and *Arabidopsis*; (**D**) Covariance analysis of TCP genes between pineapple and grape.

### 3.6. Analysis of AcTCP Promoter cis-Acting Elements

The *cis*-acting elements of the AcTCP gene promoter (upstream 2 kb region) were predicted through the PlantCARE website to understand the regulation mechanism of AcTCPs. The results showed that the promoter region contains four response elements (light, hormone, stress, and plant growth and development), with a total of 181 elements (Figure 5). Moreover, 92 light-responsive elements, 46 hormone-response elements, 25 stress-response elements, and 18 plant growth and development-related elements were found (Figure 5B). Further analysis revealed that the light-responsive elements, including AE-box, chs-CMA2a, GA-motif, GATA-motif, G-box, GT1-motif, I-box, MRE, Sp1, TCCC-motif, TCT-motif, and Box 4, were the most widely distributed among the family members, accounting for about half of all response elements. In addition, hormone-response elements, including ABRE, CGTCA-motif, GARE-motif, P-box, TATC-box, TCA-element, and TGACG-motif, were widely distributed, accounting for about one-fourth of all response elements. Box 4 accounted for the largest proportion of light-responsive elements (about 27%), and it had 10 elements on the promoter of AcTCP6. Among the phytohormone-responsive elements, ABRE was particularly abundant, accounting for about 33%, and ABRE was widely distributed on *AcTCP3*, *AcTCP5*, *AcTCP6*, *AcTCP7*, *AcTCP8*, and *AcTCP9*, with four of them on the promoter of *AcTCP4* (Figure 5C). These findings suggested that the AcTCP gene family members may be mainly regulated by a combination of light signals and multiple phytohormones.

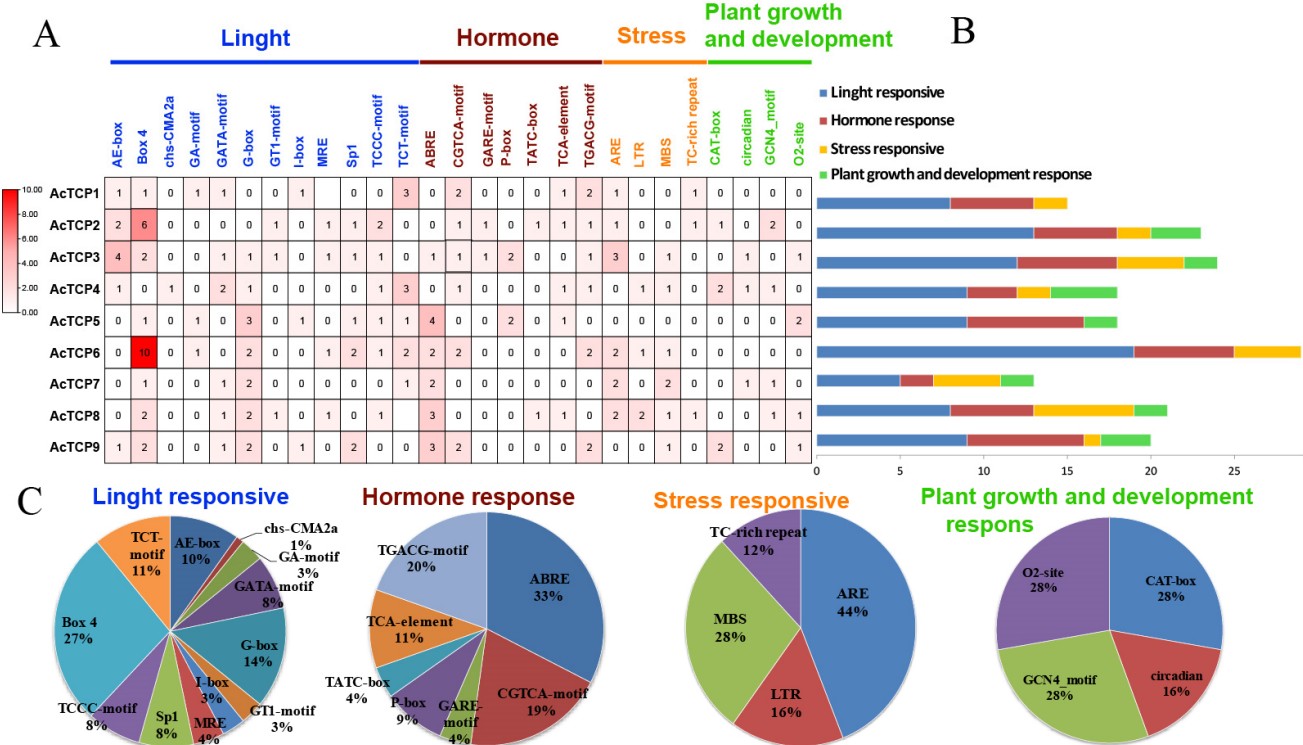

**Figure 5.** Promoter *cis*-acting elements analysis of AcTCP genes. (**A**) Analysis of *cis*-acting elements in TCP promoter region; (**B**) The number of different types of elements in TCP promoter was counted and expressed in different colors; (**C**) The pie chart shows the proportion of *cis*-acting elements in the four types of elements.

### 3.7. Expression Profile of AcTCP Genes

3.7.1. Expression Profile of AcTCP Genes during Flower Development

The expression characteristics of *AcTCPs* at different stages of flowering development were analyzed to verify the role of AcTCPs in ethylene-induced pineapple flowering development (Figure 6A). After ethylene treatment was applied for 12 h, the expression levels of nine *AcTCP* members in pineapple terminal bud increased to different degrees.

Among them, the expression levels of *AcTCP5*, *AcTCP6*, *AcTCP7*, *AcTCP8*, and *AcTCP9* increased significantly, suggesting that these AcTCP members may be involved in early floral induction in response to ethylene signals. However, the expression levels of *AcTCP6*, *AcTCP7*, and *AcTCP9* remained at a high level from 1 week after treatment to the period before flowering, indicating that these AcTCP may be involved in the differentiation process of flower organs in the late stage of pineapple flower induction.

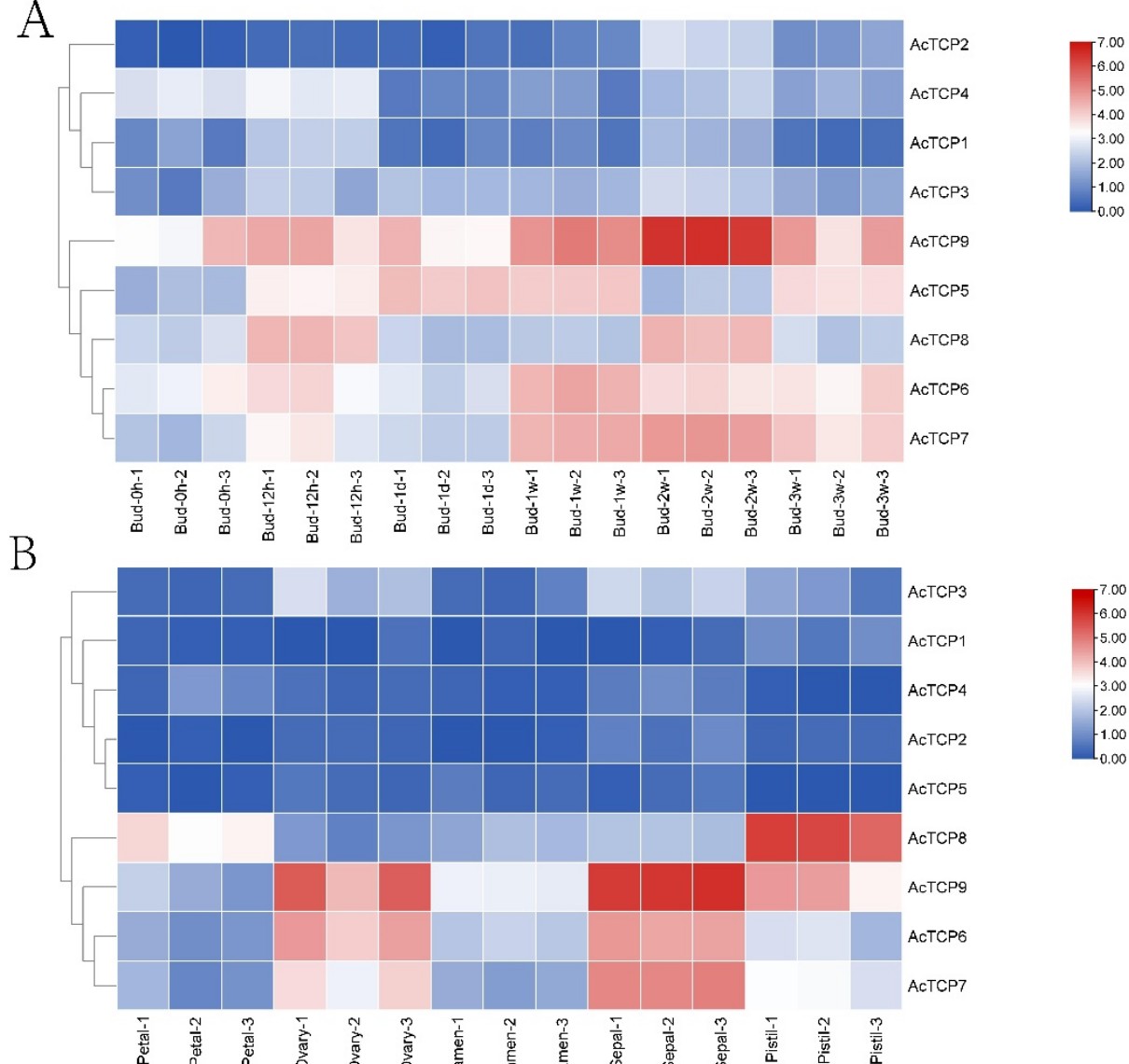

**Figure 6.** Expression profile of *AcTCP* genes in pineapple flower development: (**A**) Expression profile of *AcTCP* genes in ethylene-induced pineapple flowering development. The *x*-axis represents the different flower bud developmental stages of the pineapple; (**B**) Expression profile of *AcTCP* genes in different floral organs of pineapple. The *x*-axis represents the different flower organ developmental stages of the pineapple. The *y*-axis refers to the deduced FPKM value with Log2.

The expression patterns of *AcTCPs* in various flower organs of pineapple were examined to elucidate the role of AcTCPs in the formation of different flower organs in pineapple (Figure 6B). *AcTCP8* exhibited significant expression levels in the pistil, whereas *AcTCP7* and *AcTCP6* were highly expressed in ovary and sepal. *AcTCP9* showed high expression in ovary, sepal, and pistil. These findings indicate that distinct members of AcTCP may participate in the development of various flower organs in pineapple.

### 3.7.2. Expression Profile of AcTCPs Genes during Different Fruit Developmental Stages

The expression characteristics of *AcTCPs* at different periods of pineapple fruit were analyzed to verify the role of AcTCPs in fruit development (Figure 7). The overall trend showed that one group containing *AcTCP4*, *AcTCP8*, *AcTCP5*, *AcTCP1*, and *AcTCP2* had lower expression levels throughout fruit development, whereas the other group consisting of *AcTCP3*, *AcTCP7*, *AcTCP6*, and *AcTCP9* had higher expression levels in the early and middle fruit stage, and then these levels decreased. Therefore, AcTCP3, AcTCP7, AcTCP6, and AcTCP9 may be involved in the early development of pineapple fruit, but not in the later development and ripening process. *AcTCP3*, *AcTCP7*, *AcTCP6*, and *AcTCP9* all belong to the PCF group of class I, whereas *AcTCP4*, *AcTCP8*, and *AcTCP1* belong to the CIN group of class II, and *AcTCP5* and *AcTCP2* belong to the CYC/TB1 group of class II. In conclusion, the TCP members of the PCF group may be involved in the development of pineapple fruit, whereas members of the CIN and CYC/TB1 group are not involved in the regulation of fruit development.

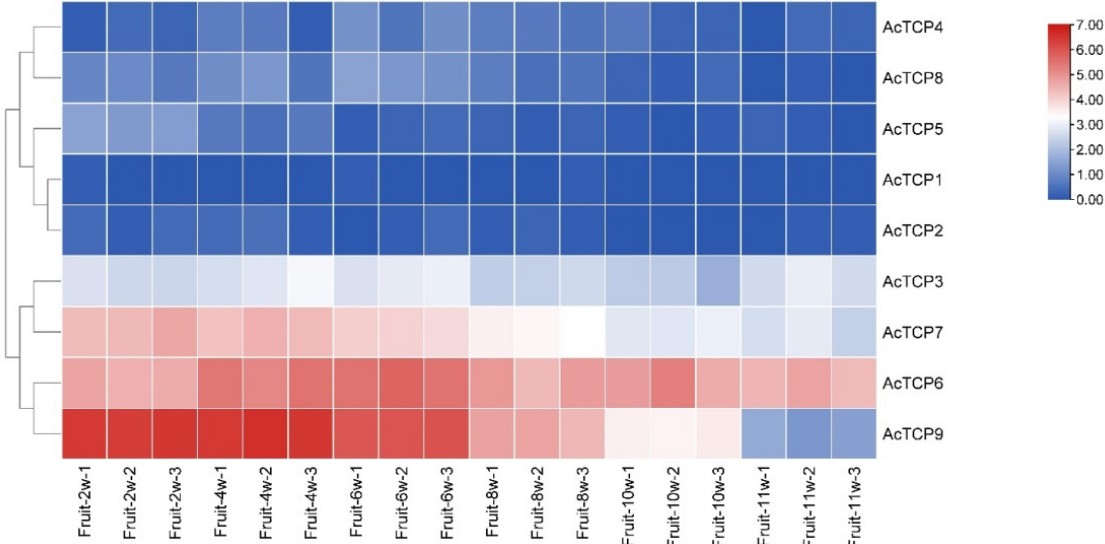

**Figure 7.** Expression profiles of AcTCP genes in fruit development of pineapple. The labels of the *x*-axis represent the different fruit developmental stages of the pineapple. The *y*-axis refers to the deduced FPKM value with Log2.

### 3.8. qRT-PCR Assays of AcTCP Expression Patterns

### 3.8.1. qRT-PCR Assays of AcTCP Expression Patterns during Flower Development

The expression characteristics of *AcTCPs* were analyzed by qRT-PCR experiments in different developmental periods of pineapple flower to further verify the roles of AcTCPs in flower development (Figure 8). The expression of *AcTCP1, AcTCP2*, and *AcTCP7* in flower buds significantly increased in the middle and late stages of ethylene treatment. By contrast, the expression of *AcTCP4* decreased significantly after ethylene treatment, and thus, AcTCP4 may play a role opposite to that of the other members during flower bud development (Figure 8A). The expression of *AcTCP4*, *AcTCP6*, *AcTCP7*, and *AcTCP9* in sepals, *AcTCP8* and *AcTCP9* in pistils, and *AcTCP5* in ovaries was significantly higher than those in other floral organs (Figure 8B). The *AcTCP* members differed greatly in terms of expression in different floral organs, and they may play a role in the development of different floral organs.

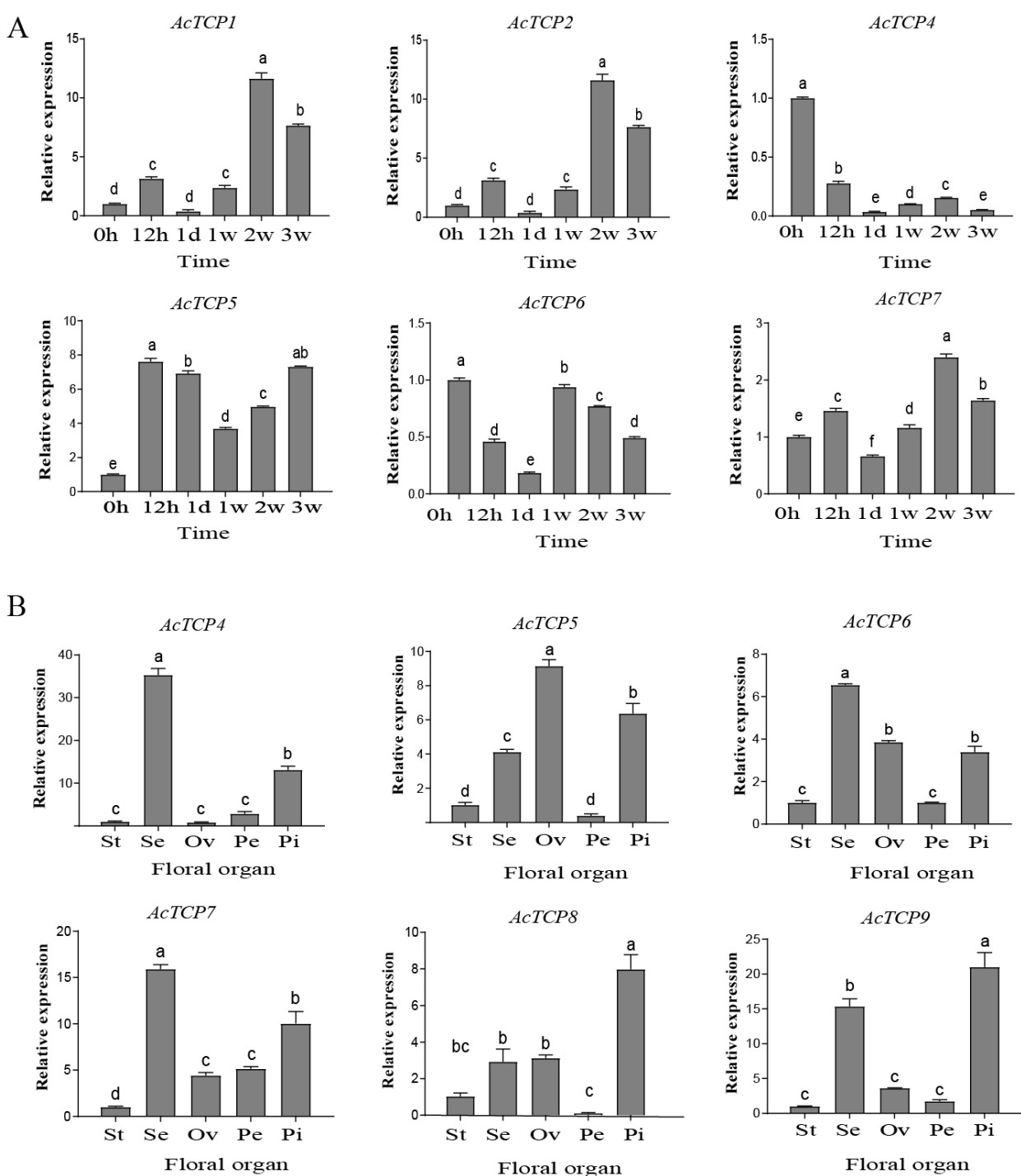

**Figure 8.** qRT-PCR assays of *AcTCP* expression patterns during flower development: (**A**) Analysis of *AcTCP* expression at six different developmental stages of ethylene-induced flower bud differentiation. The *x*-axis indicates flower bud developmental stages; (**B**) Relative expression of *AcTCP* in different floral organs, style (St), sepal (Se), ovary (Ov), petal (Pe), pistil (Pi), with style samples as reference. The *x*-axis indicates flower organs. The *y*-axis indicates the relative expression of each gene. Error bars indicate the standard deviations of the three biological replicates. Normal letters indicate significant difference at 0.01, LSD test.

3.8.2. qRT-PCR Assays of AcTCP Expression Patterns during Fruit Development

*AcTCP2*, *AcTCP4*, *AcTCP5*, *AcTCP6*, *AcTCP7*, and *AcTCP9* were all highly expressed in early fruit development and gradually degraded in later stages, suggesting that the

AcTCP family members may promote fruit ripening in early stages but are not involved in fruit softening. The expression of *AcTCP5* gradually decreased with fruit development, so AcTCP5 may inhibit fruit ripening (Figure 9). The expression patterns of most AcTCP members were consistent with the transcriptome results, indicating that the expression profiles of these genes are accurate and reliable, further confirming the validity of the experimental results. In conclusion, the expression of AcTCP family members varied greatly at different periods of flower and fruit development, and these family members may play a critical part in the development of pineapple flower and fruit.

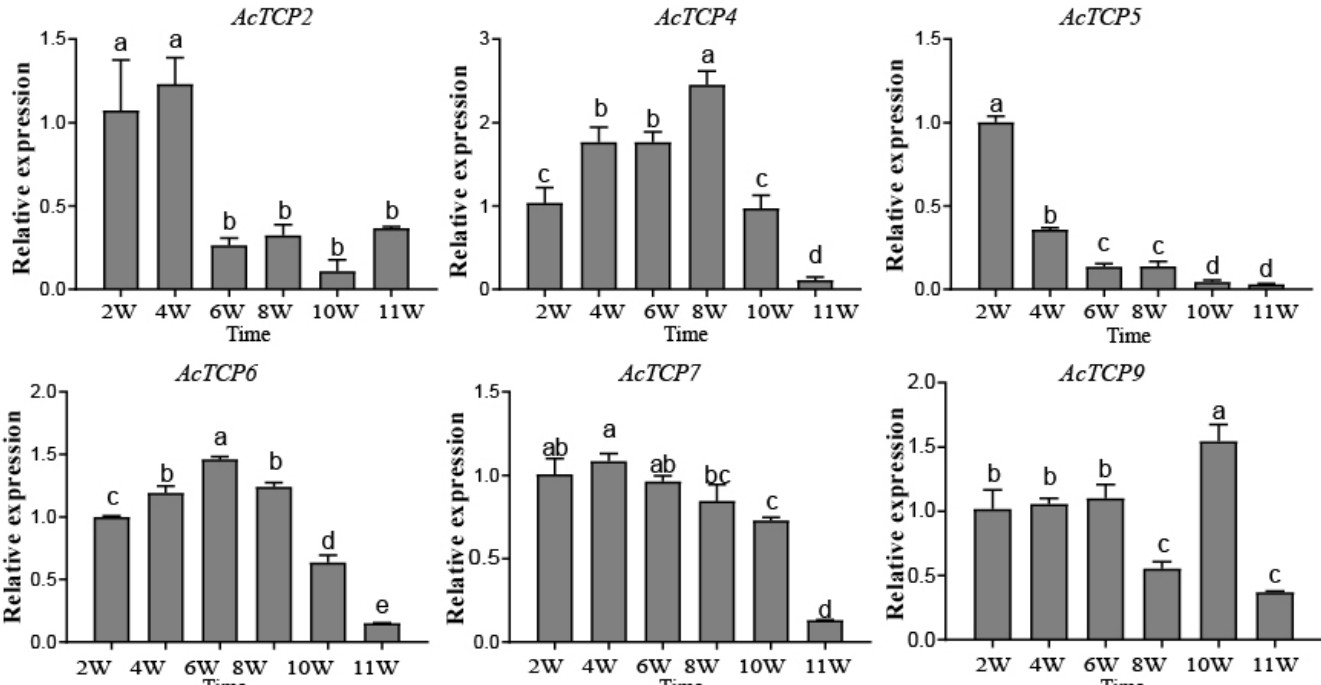

**Figure 9.** qRT-PCR assays of *AcTCP* expression in different periods of fruit development. The *x*-axis indicates fruit development 2w, Fruit development 4w, Fruit development 6w, Fruit development 8w, Fruit development 10w, Fruit development 11w. The *y*-axis indicates the relative expression of each gene. Error bars indicate the standard deviations of the three biological replicates. Normal letters indicate significant difference at 0.01, LSD test.

### 3.9. Diversified and Conserved Protein Interaction Network of AcTCP Proteins

Protein network prediction could help analyze the functions of AcTCPs and the interaction regulatory network. Here, the protein interaction prediction site (https://cn.string-db.org/, accessed on 25 September 2022) was used to predict the interaction network of nine AcTCPs (Figure 10). The results showed that AcTCP5 interacts with SPL9 protein (AT2G42200.1, Figure 10E), which is involved in the regulation of flowering and fruit development. Meanwhile, AcTCP8 may interact with FT (AT1G65480.1, Figure 10H), which is involved in the induction of flower formation. In addition, TCP proteins interact with one another, indicating that TCP members regulate the expression of downstream genes by forming polymers among themselves. These results provide an important entry point for the study of TCP transcription factors involved in pineapple flower and fruit development.

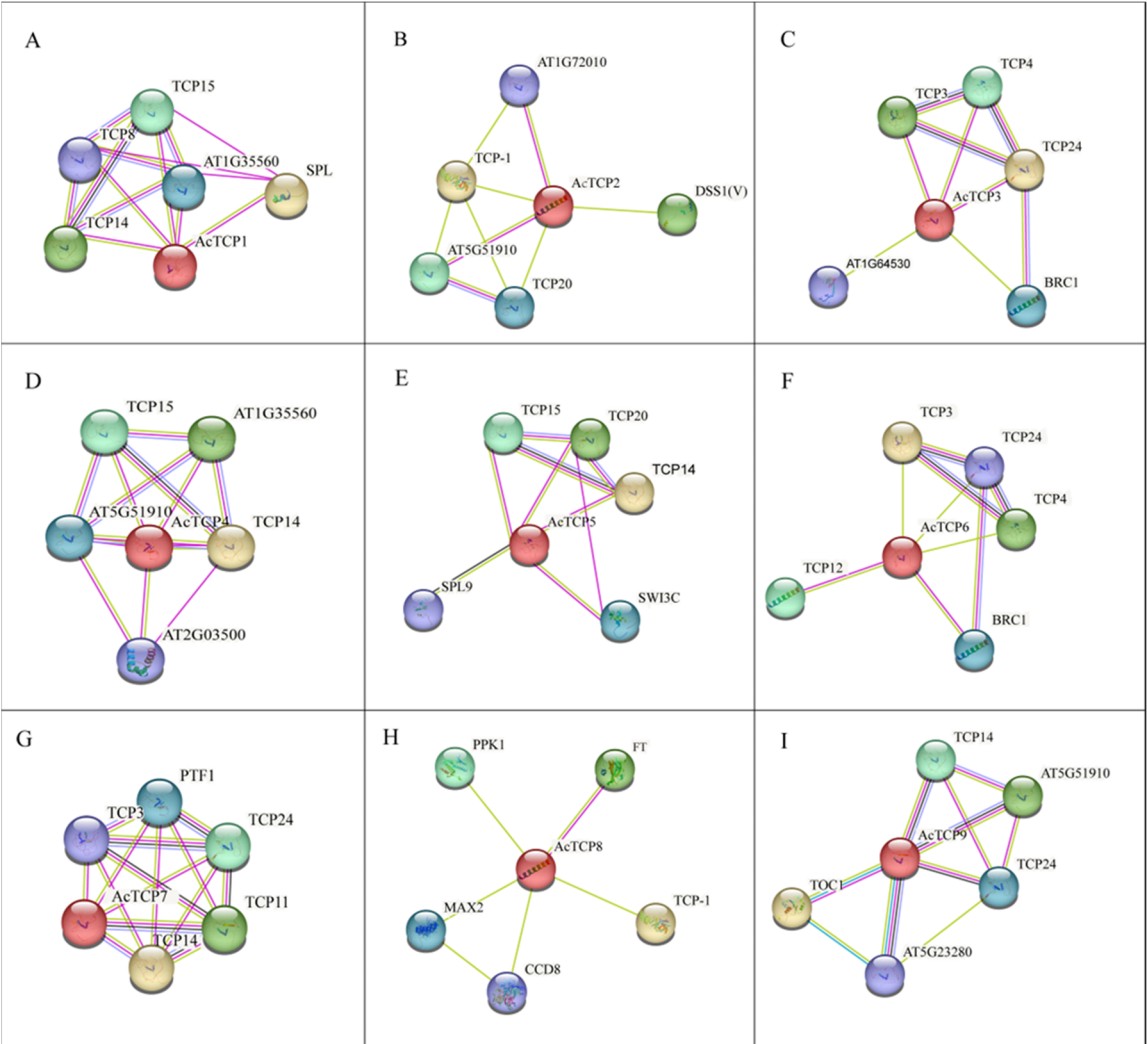

**Figure 10.** Prediction of AcTCP protein interaction regulatory network. (**A–I**) represent the network prediction of AcTCP1-AcTCP9 and its interacting proteins, respectively. The light blue line is from curated databases, the pink line is experimentally determined, the green line is the gene neighborhood, the red line is the gene fusions, the blue lines are gene co-occurrence, the yellow lines are textmining, the black lines are co-expression, and the purple lines are protein homology.

*3.10. Y2H Validation of the Interaction of AcTCP with FT and SPL*

Previous studies found that SPL and FT transcription factors regulate plant flower and fruit development [43,44], and based on protein interactions predicted that we select SPL9 (AT2G42200.1) and FT (T1G65480.1) as interaction partners (Table S3). An evolutionary tree was constructed to obtain the closest AcSPL family members (AcSPL4, AcSPL16, and AcSPL17) related to SPL9 in pineapple and the closest AcPEBP family members (AcFT2, AcFT5, and AcFT6) related to FT for yeast two-hybrid experiments to verify the results predicted by the interaction network (Figure 11). pGBKT7-AcFT2, pGBKT7-AcFT5, pGBKT7-AcFT6, pGBKT7-AcSPL4, pGBKT7-AcSPL16, pGBKT7-AcSPL17, pGADT7-AcTCP5, and pGADT7-AcTCP8 were constructed. pGBKT7-AcSPL/AcFT was co-transformed with pGADT7-empty, pGBKT7-empty with pGADT7-AcTCP, and pGBKT7-lam with pGADT7-T into AH109 yeast susceptibility as negative control. Meanwhile, pGBKT7-53 was co-transformed with pGADT7-T as positive control. The yeast cells obtained from the above co-transformation were coated on SD/Trp/leu medium for growth, and individual colonies were selected and separately coated on SD/Trp/Leu/His/Ade/3-AT to observe the colony growth. The results showed that the yeast cells from the positive

control, negative control, and experimental groups were able to grow on the SD/Trp/Leu medium, indicating that the above combination was transformed into yeast cells. Positive controls and the pGBKT7-AcSPL16 colonies co-transformed with pGADT7-AcTCP5 grew well on SD/Trp/Leu/His/Ade/3-AT (10 mM). Moreover, the pGBKT7-AcFT5 colonies co-transformed with pGADT7-AcTCP8, and the pGBKT7-AcFT6 colonies co-transformed with pGADT7-AcTCP8 grew well on SD/Trp/Leu/His/Ade/3-AT (5 mM), whereas none of the negative controls grew well. Then, 5 μL of X-α-gal was spotted on the colonies, which turned blue, indicating that the above combination could activate the expression of the downstream reporter gene. These results showed that AcTCP5 interacts with AcSPL16, and AcTCP8 interacts with AcFT5 and AcFT6 proteins.

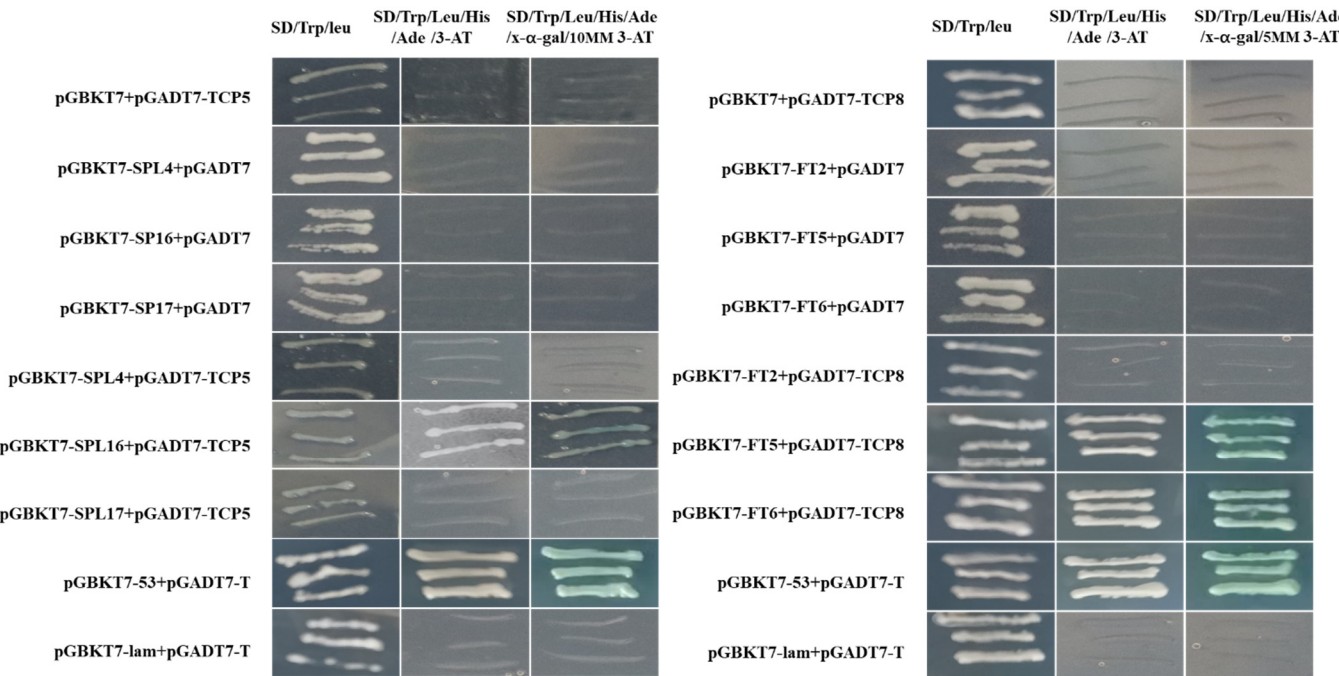

**Figure 11.** AcTCP interacted with SPL and FT in yeast two-hybrid assay. pGBKT7-lam and pGADT7-T is negative control, pGBKT7-53 and pGADT7-T is positive control. pGBKT7-AcSPL16 and pGADT7-AcTCP5 grew well in SD medium lacking Trp, Leu, His, and Ade containing 10 mM 3-AT, while the corresponding negative control did not grow normally under the same conditions. pGBKT7-AcFT5, 6 and pGADT7-AcTCP8 grew well in medium containing 5 mM 3-AT, while the corresponding negative control did not grow normally under the same conditions.

## 4. Discussion

TCP transcription factors, which are specific to plants, play a crucial role in the growth and development of plants. While the TCP gene family has been extensively studied and characterized in several plant species, there is limited systematic and comprehensive knowledge regarding the TCP gene family in pineapple. In *Arabidopsis* [7], maize [11], and grape [9], TCP members could be divided into class 1 and class 2 subfamilies and further divided into CIN, CYC/TB1, and PCF. In the present study, a similar classification exists for pineapples, as demonstrated by the results of studies on evolutionary relationship, gene structure, and amino acid sequence alignment (Figures 2 and 3). Each AcTCP member has a conserved bHLH structural domain (Figure 3C), similar to TCP members in *Arabidopsis* [7], strawberry [45], and petunia [46] species. In terms of the number of TCP members, only nine AcTCP members were identified in pineapple, far less than in *Arabidopsis* (23), rice (22), tomato (30), and apple (52) [47]. In general, the number of gene replication events is an important factor affecting the number of gene family members. No fragmental and tandem duplication events occurred in the AcTCP members in pineapple, and this phenomenon may be the main reason for the low number of TCP members in pineapple. Further

collinearity analysis between species revealed that the collinearity gene pairs of AcTCP with rice and maize were more than those with *Arabidopsis* and grapes (Figure 4), indicating that the TCP of pineapple is more closely related to the species of monocotyledon and highly conserved and long-term in evolution.

The significance of the TCP gene family in plant flowering, flower organ formation, and fruit ripening has been demonstrated in several species, such as *Arabidopsis*, chrysanthemum, and banana. The results of transcriptomic data and qRT-PCR experiments revealed that different AcTCP members have significantly different gene expression characteristics during different tissues and organs, flower development, and fruit development (Figures 6–9), indicating that the TCP gene family could participate in different processes of growth and development in pineapple. However, different TCP members have different temporal and spatial expression characteristics and functions. As members of the PCF subfamily, the expression levels of *AcTCP6*, *AcTCP7*, and *AcTCP9* significantly increased at 1 week or 2 weeks during ethylene-induced flowering (Figure 6A), and these TCP members in pineapple may perform a significant function in the flowering process of pineapple. This finding is similar to previous findings that AtTCP7 interacts with nuclear factor-Ys in *Arabidopsis* to activate the expression of *SOC1* and promote early flowering in *Arabidopsis* [14]. The expression of *AcTCP8* and *AcTCP9* was significantly higher in the pistil than in other tissues (Figure 6B), and these two genes may affect pistil development. The high expression of *PmTCP4* in incomplete flowers was found to abort pistils [48,49], and AcTCP8 is closely related to PmTCP4 and may play a similar role in pineapple. During fruit development, the expression of *AcTCP4* and *AcTCP6* tended to increase and then decrease (Figure 7), and thus, AcTCP4 and AcTCP6 may play a part in promoting fruit softening and fruit ripening. This finding is also consistent with the previous findings that the expression trend of *FvTCP9* in strawberry increased and then decreased, with the function of promoting fruit ripening [18]. In the current study, the *cis*-acting element of the promoter was analyzed to investigate the regulation of TCP gene expression, and the expression of TCP protein was found to be mainly influenced by light and hormonal signals (Figure 5). This finding is similar to previous findings, which showed that in ProTCP22::MycTCP22 plants, TCP22 formed a complex with natural CRY2 only under blue light, whereas red light not only affected the CRY2-TCP22 interaction but also induced TCP22 formation [50]. Moreover, StTCP member expression was upregulated or downregulated after MeJA treatment [51], and JA and ABA directly activate *AaTCP15* expression in response to AaGSW1 to regulate artemisinin biosynthesis [23].

Protein interaction network prediction screened SPL and FT proteins that may interact with TCP proteins and regulate flower and fruit development. For confirmation of the hypothesis, yeast two-hybrid experiments were performed to verify the interactions between pGBKT7-SPL16 and pGADT7-AcTCP5, pGBKT7-AcFT5 and pGADT7-AcTCP8, and pGBKT7-AcFT6 and pGADT7-AcTCP8 (Figure 11). Previous studies found that the FT-FD module plays a key role in the photoperiodic flowering pathway, with TCP interacting with the FT-FD complex and then binding to the downstream *AP1* promoter, enhancing its transcription and positively regulating flowering in an AP1-dependent manner [13]. SPL proteins affect plant flowering through the age pathway, and *Arabidopsis* SPL3/4/5 has also been shown to promote flowering by interacting with FD and binding to the GTAC motif on the *AP1* promoter, and delay flowering under LD after SPL3/4/5 gene knockout [52]. This study provides a reference for the regulation of plant flower and fruit development by TCP transcription factors. However, whether TCP and its interacting transcription factors (SPL and FT) can promote the expression of genes such as *SOC1*, *AP*, and *XTH*, and thus integrate into the flower- and fruit-forming pathways, needs further validation.

## 5. Conclusions

During the course of this research, a total of nine AcTCP members were successfully identified in pineapple. Their chromosomal mapping, phylogenetic, gene structure and motif, multiple-sequence alignment, covariance, promoter *cis*-acting elements, and protein

interaction prediction were systematically analyzed. The transcriptomic data and qRT-PCR analysis demonstrated an upregulation of *AcTCP5* and *AcTCP7* during the later stages of floral bud development, *AcTCP8* expression was significantly higher in the pistil than in other floral organs, and *AcTCP6* and significantly decreased in late fruit development. The yeast two-hybrid assay verified that TCP interacts with SPL and FT proteins, and some members of AcTCP may play important roles in pineapple flower and fruit development. This paper lays the foundation for further studies on the functions of the AcTCP gene family in pineapple flower and fruit development.

**Supplementary Materials:** The following supporting information can be downloaded at: https://www.mdpi.com/article/10.3390/horticulturae9070799/s1, Table S1: The table of primer for qRT-PCR. Table S2: Conserved motif sequences of the TCP family in pineapple. Table S3: Information on proteins interacting with AcTCPs among the protein regulatory networks.

**Author Contributions:** Writing—original draft preparation, Z.L.; Methodology, X.P. and Y.O.; Data curation and Software, X.Z. and R.X.; Writing—review and editing, H.Z. and Y.W.; Supervision: H.Z., Y.W. and C.W.; Conceptualization, H.Z. and Y.W.; Resources and Investigation, Z.L. and L.Z.; Funding acquisition: H.Z. All authors have read and agreed to the published version of the manuscript.

**Funding:** The project was funded by the major science and technology project of Hainan Province (ZDKJ2021014), the National Natural Science Fund of China (32160687 and 31872079), the Natural Science Foundation of Hainan Province (321RC467), and the Scientific Research Start-up Fund Project of Hainan University (KYQD-ZR-20090).

**Data Availability Statement:** The data are available on request from the corresponding author.

**Conflicts of Interest:** The authors declare that they have no conflict of interest.

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
