# Peer review of "TCP Transcription Factors in Pineapple: Genome-Wide Characterization and Expression Profile Analysis during Flower and Fruit Development"

_horticulturae, doi:10.3390/horticulturae9070799_

Round 1

Reviewer 1 Report

The paper addresses the identification and genome wide characterization of TCP transcription factors in Pineapple. Nine AcTCP genes were examined and characterized by bioinformatics approaches, to investigate the role of main regulators and hormones in the flower and fruit development. Additionally, temporal gene expression analysis of AcTCP were performed with different tissues and conditions. Also analyses of partners were performed throughout protein-protein interactions by yeast two hybrid assays. The main findings of manuscript provide information about the regulation and role of AcTCP genes during flowering and fruit development.

Overall, the manuscript is well written and structured. However, there are some points that need more clarification as highlighted below.

Introduction is ok. My suggestion is to include a topic on how transcription factors can interact with TCP proteins in regulating flower and fruit development. At this point the authors could include throughout the manuscript how the prediction of protein interaction network analysis of AcTCP proteins was based.

Material and Methods

Line 140-142. Please include additional information on how the protein interaction predictions for nine AcTCP genes and the criteria for selecting AT2G42200.1 and AT1G65480.1 as interaction partners were performed.

Results

Figure 3 A and B can be compiled into a unique figure.

What do the colors highlighted in the alignment in Figure 3B represent? Please specify.

From fig. 5 on, there has been an error in the citation of the figures. For example, in line 262, the authors cited Figure 6, and were referring to Figure 5. Please revise all figures cited in the text.

Figure 5 A and B, as designed, doesn't look good. It looks like a single figure. I suggest separating them.

Figures 5C was not cited in the topic 2.6.

Line 307. “Figure 7”

Is the pie chart (Fig 5C) showing the proportion of cis-acting elements in the four element types correct? The proportion of the light responsive elements represents 97%, hormones 86% and plant growth and development 50%. Only stress response represents 100%. Provide more information about this analysis.

In the heatmap graphs, 5, 6 and 7, what do the scales (0-10) Fig. 5 and (0-7) Figs. 6 and 7 mean?

Line 326-327 "The expression of AcTCP1, AcTCP2, AcTCP5, AcTCP6, and AcTCP7 in flower buds significantly increased in the middle and late stages of ethylene treatment and remained highly expressed in the late stages." This is not in accordance with Fig. 8, AcTCP6 in flower buds significantly reduced in the middle (Bud1d) and late stages (Bud3W) of ethylene treatment.

In the legend of Fig8. Include the meaning of the abbreviations St. Se, Ov, Pe, Pi and Bud.

In the Fig.9. Include the meaning of Pu 2, 4, 6, 8, 10, 11.

Prediction of protein networks (line 355-358) indicated AcTCP5 and SPL9 (AT2G42200.1), and AcTCP8 (CYC/TB1) and FT (AT1G65480.1) as partners respectively. What explains the interaction analyses (Fig. 11) being conducted with AcTCP9 (PCF). They are of different TCP subgroups.

Also with regard to protein-protein interaction assays, the authors could include the orthologs identified in A. thaliana, SPL9 (AT2G42200.1) and FT (AT1G65480.1) in the 2YH assays, and test the interaction with AcTCP5 and AcTCP8, respectively to see if these regulatory mechanisms are conserved in plants and between different species.   

Why in the studied interactions were not performed with the cloned constructs in opposite orientations, for example "pGBKT7-TCP5 + pGADT7SPL16"? My questioning is because transcription factors usually have the ability to bind to DNA and transactivate gene expression.

Line 387-389. The conclusion of this topic is a bit confusing, and suggest that AcTCP5 interacts with AcSPL16, FT5 and FT6.

The legend in figure 11 should be improved by including all information about vectors, partners, culture media, and 3AT concentrations.

Discussion

Line 404. “Figure 3”

Please review all figures cited in the discussion.

Conclusion

The conclusion is appropriate and in accordance with the main findings of the study

Overall the quality of English language is fine.

Author Response

Response to comments (Reviewer1)

  1. Comment: Introduction is ok. My suggestion is to include a topic on how transcription factors can interact with TCP proteins in regulating flower and fruit development. At this point the authors could include throughout the manuscript how the prediction of protein interaction network analysis of AcTCP proteins was based.

Response: According to your suggestion, we have added the contents related to TCP regulation of flower and fruit development in introduction. These findings relate to the mechanisms by which TCP proteins interact with other proteins to regulate downstream genes.

  1. Comment: Line 140-142. Please include additional information on how the protein interaction predictions for nine AcTCP genes and the criteria for selecting AT2G42200.1 and AT1G65480.1 as interaction partners were performed.

Response: According to your suggestion, we have added the interaction relationship prediction of nine AcTCP proteins in the attachment.

It has been widely proved that SPL and FT participate in the flower formation of plants, and our previous study also found that AcSPL and AcFT participate in the flower formation of pineapple. Therefore, we selected the most closely related members of the pineapple with SPL9 (AT2G42200.1) and FT (AT1G65480.1) for TCP interaction analysis.

  1. Comment: Figure 3 A and B can be compiled into a unique figure.

Response: We have merged the original Figure 3A and 3B into Figure 3A according to your suggestion.

  1. Comment: What do the colors highlighted in the alignment in Figure 3B represent? Please specify.

Response: The colors highlighted in the alignment in Figure 3B represent conserved motif sequences of the AcTCP family. The detailed sequences are shown in Supplemental Table S2.

  1. Comment: From Fig. 5 on, there has been an error in the citation of the figures. For example, in line 262, the authors cited Figure 6, and were referring to Figure 5. Please revise all figures cited in the text.

Response: We have carefully checked and revised all figures cited in the full text.

  1. Comment: Figure 5 A and B, as designed, doesn't look good. It looks like a single figure. I suggest separating them. Figures 5C was not cited in the topic 2.6.

Response: Your suggestion is very good. However, since the information in Figure 5A and Figure 5B is one-to-one correspondence, it is more intuitive to put it closer. This kind of diagram is also used in this part of other papers (Zhang et al. 2021, Wang et al. 2022). Therefore, we have retained the previous layout and hope to get your approval.

In addition, we have added labels to the sections of topic 2.6 referenced in Figure 5B and 5C.

Zhang et al. 2021. Genome-Wide identification of JRL Genes in Moso Bamboo and their expression profiles in response to multiple hormones and abiotic stresses. Frontiers in Plant Science. https://doi.org/10.3389/fpls.2021.809666

Wang et al. 2022. Identification and characterization of trihelix transcription factors and expression changes during flower development in pineapple. horticulturae. https://doi.org/10.3390/horticulturae8100894

  1. Comment: Line 307. “Figure 7” Is the pie chart (Fig 5C) showing the proportion of cis-acting elements in the four element types correct? The proportion of the light responsive elements represents 97%, hormones 86% and plant growth and development 50%. Only stress response represents 100%. Provide more information about this analysis.

Response: Sorry for the error in data processing. We have re-plotted the graph and the new pie chart is 100%.

  1. Comment: In the heatmap graphs, 5, 6 and 7, what do the scales (0-10) Fig. 5 and (0-7) Figs. 6 and 7 mean?

Response: We have added "The Y-axis refers to the deduced FPKM value with Log2." to explain the scale.

  1. Comment: Line 326-327 "The expression of AcTCP1, AcTCP2, AcTCP5, AcTCP6, and AcTCP7 in flower buds significantly increased in the middle and late stages of ethylene treatment and remained highly expressed in the late stages." This is not in accordance with Fig. 8, AcTCP6 in flower buds significantly reduced in the middle (Bud1d) and late stages (Bud3W) of ethylene treatment.

Response: Thanks for your correction. We have modified the statement to “The expression of AcTCP1, AcTCP2, and AcTCP7 in flower buds significantly increased in the middle and late stages of ethylene treatment”.

  1. Comment: In the legend of Fig8. Include the meaning of the abbreviations St. Se, Ov, Pe, Pi and Bud.

Response: We have added relevant information and modified the legend to “(B) Relative expression of AcTCP in different floral organs, style(St), sepal(Se), ovary(Ov), petal(Pe), pistil(Pi), with style samples as reference.”

  1. Comment: In the Fig.9. Include the meaning of Pu 2, 4, 6, 8, 10, 11.

Response: We have modified the information about the x-axis in the picture and added to the comment message “The x-axis indicates fruit development 2w, Fruit development 4w, Fruit development 6w, Fruit development 8w, Fruit development 10w, Fruit development 11w. The y-axis indicates the relative expression of each gene.”

  1. Comment: Prediction of protein networks (line 355-358) indicated AcTCP5 and SPL9 (AT2G42200.1), and AcTCP8 (CYC/TB1) and FT (AT1G65480.1) as partners respectively. What explains the interaction analyses (Fig. 11) being conducted with AcTCP9 (PCF). They are of different TCP subgroups.

Response: To make this paper seem more logical, we have removed the experimental results of AcTCP9 interaction and added the results of AcTCP8 interaction with AcFT protein.

  1. Comment: Also with regard to protein-protein interaction assays, the authors could include the orthologs identified in A. thaliana, SPL9 (AT2G42200.1) and FT (AT1G65480.1) in the 2YH assays, and test the interaction with AcTCP5 and AcTCP8, respectively to see if these regulatory mechanisms are conserved in plants and between different species.  Why in the studied interactions were not performed with the cloned constructs in opposite orientations, for example "pGBKT7-TCP5 + pGADT7SPL16"? My questioning is because transcription factors usually have the ability to bind to DNA and transactivate gene expression. 

Response: That's very good advice. The main aim of our study was to clarify the interaction between genes that regulate pineapple flower formation, so Arabidopsis SPL9 (AT2G42200-1) and FT (AT1G65480-1) were not included in the Y2H experiment at the time. However, we selected the closest homologous gene SPL9 (AT2G42200-1) and FT (AT1G65480-1) in pineapple, and tested their interaction with AcTCP5 and AcTCP8, respectively.

Due to the limited revision time of the paper, we did not have time to construct vectors and Y2H experiment. Follow-up experiments could compare these results to see if this regulatory mechanism is conserved across species. The same problem exists in the verification of inverse interaction relations. In the follow-up research work, we will further systematically verify the transcriptional regulatory mechanism and functional verification.

  1. Comment: Line 387-389. The conclusion of this topic is a bit confusing, and suggest that AcTCP5 interacts with AcSPL16, FT5 and FT6.

Response: According to your suggestion, we have revised the description of the conclusion. "These results showed that AcTCP5 interacts with AcSPL16, AcTCP8 interacts with AcFT5 and AcFT6 proteins".

  1. Comment: The legend in figure 11 should be improved by including all information about vectors, partners, culture media, and 3AT concentrations.

Response: According to your suggestion, we have added a detailed description of the legend in Figure 11. “pGBKT7-lam and pGADT7-T was as negative control, pGBKT7-53 and pGADT7-T was as positive control. pGBKT7-AcSPL16 and pGADT7-AcTCP5 grew well in SD medium lacking Trp, Leu, His, and Ade containing 10 mM 3-AT, while the corresponding negative control did not grow normally under the same conditions. pGBKT7-AcFT5, 6 and pGADT7-AcTCP8 grew well in medium containing 5 mM 3-AT, while the corresponding negative control did not grow normally under the same conditions.”

  1. Comment: Line 404. “Figure 3” Please review all figures cited in the discussion.

Response: Thanks for your correction. We have carefully checked and revised all figures cited in the discussion.

Reviewer 2 Report

Authors have done a thorough examination of the TCP family in pineapple was done in the current study since it has not been previously published and because the TCP family is known to play a significant role in flower and fruit development. They have done some series of extensive bioinformatic analyses such as chromosome mapping, phylogenetics, gene structure and motif, multiple-sequence alignment, covariance, and promoter cis-acting elements of nine non-redundant TCP genes. Authors also done the experimental analysis to support this finding. This study is the groundwork for functional validation of pineapple genes was laid by additional analysis of the expression patterns of the AcTCP genes at various stages of flower and fruit development and by the prediction and validation of the proteins that interact with TCP.

However, the manuscript still needs improvisation. Therefore, I recommend the authors to incorporate the following points in the manuscript for further consideration.

Check the space and punctuation errors throughout the manuscript. Eg. Lines 41

Authors must concentrate on the formatting, and use of symbols, etc. in throughout manuscript.

Gene names should be in italics. Check throughout the manuscript and fix it.

Check and edit it if the words start with capital letters in some spots in the midst of the line. Eg. Lines: 41, 117

Scientific names should be full form in first mention and others are abbreviated form. Eg. Ananas comosus and A. comosus, (Check thoroughly).

Line 122: Uniport à UniProt

Please include the source URL whenever the database name is used, notably in the materials and methods section. Readers will benefit from it.

Section 2.7 is not clear. revise it with detailed protocol.

Please describe more about in the RNA extraction section in the materials and method section.

How much quantity of sample used for RNA isolation. What is the intensity and concentration of RNA?

Why did authors specifically use 5μg of total mRNA for cDNA synthesis instead of 1μg.

Figure 5C, 11 seems blurred, enhance the resolution.

Gene names in figure 8 and 9 should be italicized, and the figure description should include information about the error bars and various letters.

Figure 8 and 9 should be in color.

Figures can be enlarged in the manuscript. It will be useful to the readers.

A separate statistical analysis section needs to be provided in the materials and methods section.

Discussion needs to be improved.

Authors should write a few lines about the future perspectives or hypothesize about the study in the conclusion section.

Some minor corrections were highlighted in the manuscript PDF file.

Check the space and punctuation errors throughout the manuscript.

Author Response

Response to comments (Reviewer 2)

  1. Comment: However, the manuscript still needs improvisation. Therefore, I recommend the authors to incorporate the following points in the manuscript for further consideration. Check the space and punctuation errors throughout the manuscript. Eg. Lines 41. Authors must concentrate on the formatting, and use of symbols, etc. in throughout manuscript.

Response: Thanks for your correction. We have carefully checked the space and punctuation errors throughout the manuscript.

  1. Comment: Gene names should be in italics. Check throughout the manuscript and fix it.

Response: We have carefully reviewed the writing of all the gene names in this manuscript and revised them.

  1. Comment: Check and edit it if the words start with capital letters in some spots in the midst of the line. Eg. Lines: 41, 117

Response: We have carefully reviewed and corrected these errors.

  1. Comment: Scientific names should be full form in first mention and others are abbreviated form. Eg. Ananas comosusand A. comosus, (Check thoroughly).

Response: We have carefully reviewed and corrected related description.

  1. Comment: Line 122: Please include the source URL whenever the database name is used, notably in the materials and methods section. Readers will benefit from it.

Response: Thanks for your suggestion. We have added the related source URL in the text.

  1. Comment: Section 2.7 is not clear. revise it with detailed protocol. Please describe more about in the RNA extraction section in the materials and method section. How much quantity of sample used for RNA isolation. What is the intensity and concentration of RNA? Why did authors specifically use 5μg of total mRNA for cDNA synthesis instead of 1μg.

Response: Thanks for your suggestion. We have added the detailed description of RNA extraction in the materials and method section.

Each sample was weighed with 500 mg of pineapple tissue for total RNA extraction using the RNA extraction kit (Huayueyang, China). RNA quality and concentration were detected using NanoDrop™ One/OneC Spectrophotometer (Thermo Fisher Scientific, Waltham, MA, USA) and 1.5% agarose gels. The concentrations of obtained total RNAs were 150-500 ng/μL. Then the first-strand cDNAs were synthesized from 1 μg of total RNA via a Revert Aid First-Strand cDNA Synthesis Kit (Thermo Fisher Scientific, USA).

  1. Comment: Figure 5C, 11 seems blurred, enhance the resolution.

Response: We have changed the color scheme and resolution of the images to make them clearer.

  1. Comment: Gene names in figure 8 and 9 should be italicized, and the figure description should include information about the error bars and various letters.

Response: We have corrected these errors and added relevant information according to your suggestion.

  1. Comment: Figure 8 and 9 should be in color. Figures can be enlarged in the manuscript. It will be useful to the readers.

Response: We have modified the colors of the diagram and enlarged the image according to your suggestions.

  1. Comment: A separate statistical analysis section needs to be provided in the materials and methods section.

Response: We supplemented the separate statistical analysis method in the experimental methods section.

  1. Comment: Discussion needs to be improved. Authors should write a few lines about the future perspectives or hypothesize about the study in the conclusion section.

Response: Thanks for your suggestion. We have added the relevant description in the conclusion section.

  1. Comment: Some minor corrections were highlighted in the manuscript PDF file.

Response: We have modified these errors according to your suggestions.

Round 2

Reviewer 1 Report

In this revised version the authors present additional data, enriched the Introduction, Material and Methods, Results, Discussion and Conclusion and made improvements throughout the document. In this sense the authors responded my initial concerns. 

Reviewer 2 Report

I endorse the manuscript for publication